# Eosinophilic Granulomatosis with Polyangiitis Relapse after COVID-19 Vaccination: A Case Report

**DOI:** 10.3390/vaccines10010013

**Published:** 2021-12-23

**Authors:** Giulia Costanzo, Andrea Giovanni Ledda, Alessandra Ghisu, Matteo Vacca, Davide Firinu, Stefano Del Giacco

**Affiliations:** Department of Medical Sciences and Public Health, University of Cagliari, 09131 Cagliari, Italy; andre.giovanni.ledda@gmail.com (A.G.L.); alessandraghisu@yahoo.it (A.G.); vacca.matte@gmail.com (M.V.); davide.firinu@unica.it (D.F.); delgiac@gmail.com (S.D.G.)

**Keywords:** EGPA, COVID-19, vaccine, mRNA BNT162b2 vaccine, relapse

## Abstract

Background: We here describe the case of a 71-year-old Caucasian woman previously diagnosed with Eosinophilic Granulomatosis with Polyangiitis (EGPA) that had been treated with Mepolizumab, an anti-IL5 monoclonal antibody, since 2019 with a good clinical response. Methods: She had a mild COVID-19 in December 2020 and she tested negative for SARS-CoV-2 infection in only late January 2021. In April 2021 she received the first dose of mRNA BNT162b2 vaccine. Ten days later she developed myalgia, dyspnea and numbness of the limbs due to a relapse of EGPA that occurred during Mepolizumab treatment.

## 1. Introduction

The ongoing COVID-19 pandemic is being defined also by an unprecedented vaccine program with the innovative use of mRNA vaccines. Vaccination represents the only reliable means of quickly mitigating the spread and impact of COVID-19 in the near future. Patients affected by immune-mediated inflammatory disorders (IMID) are encouraged to be vaccinated because of their higher risk for both the altered regulation of the immune system itself and for the immunosuppressive effects of therapies.

Viral infections may induce autoantibody production as well as overt autoimmune disease: this has been observed for EBV, HIV, HBV and also SARS-CoV-2, among others. Although there are scant data about the capacity of SARS-CoV-2 to induce autoimmunity, there is evidence that in the sera of severely ill patients with COVID-19 high titers of autoantibodies (ANA, lupus anticoagulant, ANCA) may be detected [1]. In one case, an Italian research group described a flare up of Systemic Sclerosis after severe respiratory infection by COVID-19 [2]. It is known that in patients with an established IMID the vaccine may, rarely, act as a trigger of disease relapse, and, in patients prone to developing an autoimmune disease, it may shift the balance towards self-reaction as well [3].

## 2. Materials and Methods

Here we describe the peculiar medical history of a 71-year-old woman who was previously diagnosed with EGPA and was being treated with subcutaneous Mepolizumab, 100 mg per month; Tiotropium Bromide, 2.5 mcg 2 puffs per day; Budesonide + Formoterol, 160/4.5 mcg 2 times per day; Atorvastatin, 20 mg once a day and Omeprazole, 20 mg/day.

Her medical history began in 2014, when she was diagnosed with asthma. In 2015, following the onset of polymyalgia and polyarthralgia, she carried out diagnostic investigations that led to a diagnosis of EGPA based on Severe Eosinophilic Asthma, Chronic Rhinosinusitis and a blood eosinophil count of 0.86 × 10^3^ mcL and p-ANCA positive test. After corticosteroid therapy she achieved remission of vasculitic manifestations but poorly controlled asthma persisted over years with a mean asthma control test (ACT) of 12 (ACT lower than 20 means poor controlled asthma, ACT equals or highest than 20 means good controlled asthma. ACT of 25 is the best score that means optimum asthma control).

She had frequent asthma exacerbations and respiratory infections, for which, over time, she underwent several courses of systemic oral corticosteroids (OCS). We evaluated the patient in 2019 and we started Mepolizumab, 100 mg subcutaneously per month. She achieved good asthma control (ACT of 22) and managed to persistently suspend the OCS with no more exacerbations requiring OCS. At the time of the last evaluation in 2020 peripheral blood eosinophils were 0.16 × 10^3^/mcL, 2.6%, c-ANCA and p-ANCA were both negative and a total IgE 244.00 KU/L (normal range < 100 KU/L). On spirometry, she reported moderate to severe obstructive deficit. The patient attended at our clinic until December 2020, when she was diagnosed with SARS-CoV-2 infection. During the infection she stayed at home, reporting mild dyspnea and low-grade fever for a few days associated with arthromyalgia while mepolizumab was continued.

At the end of January 2021 she was declared clinically recovered and in April 2021, 3 months after clinical recovery, she was given a single dose of the BNT162b2 mRNA vaccine for SARS-CoV-2. At that time her p-ANCA titer was undetectable and a recent CT scan performed after COVID-19 was negative for lung lesions. After 10 days, she came to our clinic for the monthly administration of Mepolizumab complaining of increasing myalgia and polyarthralgia that started just after vaccine administration. Under the suspicion of statin-associated myopathy atorvastatin was suspended and complete blood counts and CPK were requested. During this period, she did not take any drugs or supplements. She quickly reported worsening symptoms with walking limitations, paresthesia of the right upper and left lower limb, dyspnea on mild exertion and cough associated with chest pain. There was an increase in Creatine phosphokinase (955 U/L) (normal range 34–145 U/L) and hypereosinophilia (4.3 × 10^3^/mcL, 29.4% normal value 0.1–0.5 × 10^3^/mcL) and she was admitted to our hospital. SARS-CoV-2 rt-PCR on rhinopharyngeal (RP) swab was repeatedly negative (Table 1).

The chest CT scan showed areas of parenchymal thickening attributable to pneumonia and areas of ground glass organization indicative of interstitial pneumonia. During the hospitalization, the patient complained of the persistence of the paresthesia of the first three fingers of the right hand associated with moderate lack of strength in opposition and of the plantar and lateral region of the right foot.

The inflammatory markers C-reactive protein (CRP) and erythrocyte sedimentation rate (ESR) were respectively 79 mg/dL and 80 mm/h (n.v: CRP 0-0.5, ESR 0-29), p-ANCA value was 130 UI/mL (normal range < 1.4 UI/mL). Her Birmingham Vasculitis Activity Score (BVAS) score was 11/63 (normal score 0/63).

She received pulsed intravenous corticosteroid treatment with 250 mg of methylprednisolone for three days, then switched to oral 1 mg/kg/day of prednisone. The patient gradually recovered from myositis and the blood eosinophil counts were dramatically reduced, prednisone was tapered until suspension after six months. She was discharged after 12 days of hospitalization.

## 3. Results

She received pulsed intravenous corticosteroid treatment with 250 mg of methylprednisolone for three days, then switched to oral 1 mg/kg/day of prednisone. The patient gradually recovered from myositis and the blood eosinophil counts were dramatically reduced, prednisone was tapered until suspension after six months.

## 4. Discussion

Our patient had an exacerbation of pre-existent EGPA occurring after SARS-CoV2- vaccination. In this case, the exacerbation might be linked to the activation of an immune response related to a previous SARS-CoV-2 infection, which occurred with a mild course, and by the subsequent administration of the first dose of the mRNA BNT162b2 vaccine. At the time of admission, the patient presented with dyspnea, arthtromyalgia and mononeuritis multiplex consistent with an exacerbation of EGPA, with a close temporal relationship to vaccination.

Indeed, a recent paper reported a description of immune-mediated disease flare or new-onset temporally associated with SARS-CoV-2 vaccination. They found 27 cases among five large cohorts from Israel, the UK and the USA. They reported three cases of de novo onset of vasculitis. None of them had positive ANCA antibodies [4]. In addition, a group of 7 cases of acute demyelinating disease of the CNS have been described following COVID-19 vaccination [5].

In some cases, vaccines could overstimulate the immune-mediated pathways that exert a role in disease pathophysiology, even in patients treated with targeted monoclonal antibodies. Previously, to clarify and investigate the link between the vaccine and immune-mediated disease, Gunes et al. evaluated the possible association between influenza vaccination and the onset or relapse of psoriasis [6]. Their data suggested that the influenza H1N1 vaccine used could have a relationship with developing psoriasis. A more recent paper discusses the role of vaccination in acute immune-mediated disorders of the central nervous system [7]. Based on their findings, there are no convincing reports of vaccines triggering a relapse of an existing condition.

Adjuvants are sometimes considered the culprit of the onset or exacerbation of immune mediated diseases. This opinion is supported by the recognition and description of Gulf War Syndrome [8].

A recent case report described the onset of ANCA-associated glomerulonephritis after the Moderna vaccine [9]. In the literature there are reports of IgA nephropathy after vaccination for SARS-CoV-2, especially after the second dose, that may be linked to the strong capability of the mRNA vaccines to induce both cellular and humoral immune responses [10].

Our patient presented an exacerbation of pre-existent EGPA with evidence of eosinophilia and p-ANCA positivity and organ involvement that started 10 days after vaccination for SARS-CoV-2. Of note, the exacerbation occurred despite the patient continuing to take anti-IL5 therapy (Mepolizumab). She also had mild COVID-19 three months before the vaccine and we think this may have had an impact on the clinical course. In the recent literature there are reports of new onset of ANCA vasculitis but with a predominantly renal impairment in term of crescentic glomerulonephritis [10,11]. As far as we know, this is the first case report in literature of EGPA flared after the first dose of the mRNA BNT162b2 vaccine. The timeline of the clinical appearance (Figure 1) of symptoms after vaccination supports the idea of the delicate balance of immune homeostasis in such cases being momentarily shifted to a pro-inflammatory state by vaccination. This idea is supported also by another case report of vasculitis, Henoch–Schönlein Purpura, following the first dose of COVID-19 Viral Vector Vaccine [12].

In the literature there are only anecdotal reports of IMID flares after COVID-19 vaccination and given the large number of patients who have been vaccinated to date [13], it is remarkable that there are very few reports of individuals experiencing new inflammatory disease activity after vaccination [14], including patients naïve to or with previous SARS-CoV-infection [15]. Large prospective controlled studies and data from registries are required to establish a possible relationship between COVID-19 vaccines and new onset immune mediated disorders or relapse in patients with previous SARS-CoV-2 infection. At the present time, we think the benefits of COVID-19 vaccination clearly outweigh the potential risks also among subjects affected by IMID.

In addition, the described relapse of eosinophilic vasculitis after a single dose of mRNA BNT162b2 vaccine may be also linked to the resolution of COVID-19 just three full months before the vaccination. Perhaps it would have been prudent to delay the first dose of mRNA BNT162b2 vaccine for at least 5 months from full recovery. This could have reduced the probability of eliciting a massive immune system response to such an immunogenic vaccine while also securely giving to the patient vaccination, the only realistic protection to SARS-CoV-2 infection.

## Figures and Tables

**Figure 1 vaccines-10-00013-f001:**
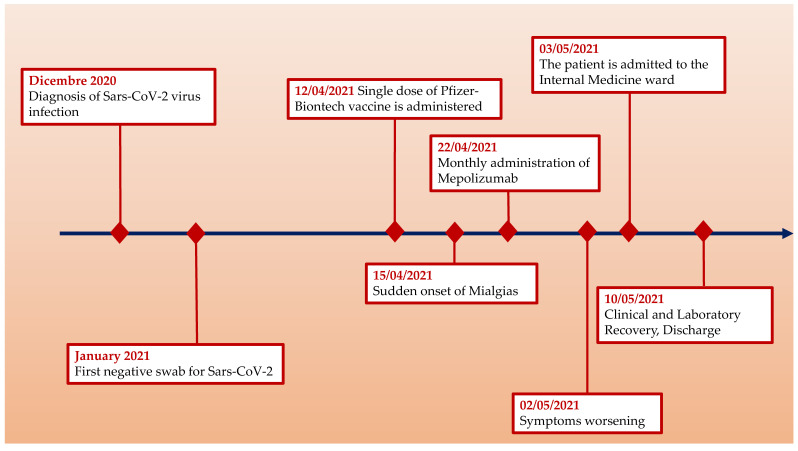
Clinical timeline.

**Table 1 vaccines-10-00013-t001:** Principal clinical features before and after Vaccine administration.

Laboratory and Instrumental Tests	Last Check before Vaccine Administration	First Check Upon Entering the Ward
CPK	110 U/L	955 U/L
Eosinophil Count	0.16 × 10^3^ (2.6%)	4.3 × 10^3^ (29.4%)
p-ANCA	NEGATIVE	130 UI/mL
Chest CT Scan	NEGATIVE	Parenchymal thickening, areas of ground glass

## Data Availability

The data used to support the findings of this study are available from the corresponding author upon request.

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
