# Peer review of "Eosinophilic Granulomatosis with Polyangiitis Relapse after COVID-19 Vaccination: A Case Report"

_vaccines, 2021, doi:10.3390/vaccines10010013_

Round 1

Reviewer 1 Report

Costanzo et al present an interesting case. However some changes are to be considered before publication of the article:

-EGPA is not a standard abbreviation and should therefore be omitted from the title.

-line 32: relapse of ... . Please be specific to avoid confusion

-Please provide normal values/reference values for the ACT

-Please provide normal values for eosinophil counts in your lab

-Same for IgE

-Typo line 67 "??"

-Please also provide normal values for CPK

-And for p-Anca

-CPK is not a standard abbreviation and should be defined the first time used in this manuscript

-Same for BVAS. Plus reference range for BVAS

-How long was the patient hospitalised?

-The authors certainly describe an interesting case that touches a timely nature. However, I am missing a key clinical message that describes a learning/teaching point and may be of interest to the readership/generalizeable and transducable to clinical care. Adding a paragraph (+discussion) would clearly improve the value of the case  and the manuscript. i.e. what is the conclusion?

Author Response

1. Costanzo et al present an interesting case. However some changes are to be considered before publication of the article:EGPA is not a standard abbreviation and should therefore be omitted from the title.

We thank the reviewer for her/his comment. We add omitted the abbreviation from the title.

2. line 32: relapse of ... . Please be specific to avoid confusion

We thank the reviewer for her/his suggestion. We modify the text

3. Please provide normal values/reference values for the ACT

We thank the reviewer for her/his suggestion, we add the normal values for ACT.

4. Please provide normal values for eosinophil counts in your lab

We add the normal values for eosinophil counts.

5. Same for IgE

We add the normal values for IgE.

6. Typo line 67 "??"

We apologize for the orthographic mistake. We erase the double question marks.

7. Please also provide normal values for CPK

We add the normal values for CPK.

8. And for p-Anca

We add the normal values for p ANCA

9. CPK is not a standard abbreviation and should be defined the first time used in this manuscript

We thank the reviewer for rainsing her/his note. We amended the text in agreement with the useful suggestions.  

10. Same for BVAS. Plus reference range for BVAS

Thank the reviewer for her/his note. We amended the text in agreement with the useful suggestions

11. How long was the patient hospitalised?

The patient was discharged after 12 days of hospitalization. We add this information to the text.

12. The authors certainly describe an interesting case that touches a timely nature. However, I am missing a key clinical message that describes a learning/teaching point and may be of interest to the readership/generalizeable and transducable to clinical care. Adding a paragraph (+discussion) would clearly improve the value of the case  and the manuscript. i.e. what is the conclusion?

We kindly thank the reviewer for her/his useful notes. We implement the discussion paragraph in the manuscript.

Reviewer 2 Report

In this cae report, the author present a patient exhibiting EGPA relapse after Sars-Cov2 vaccination. She previously undewent Covid-19 without clinical  effects. The patient after vaccination was positive for ANCA. Did the patient exhibited anti-IFNalpha antibodies?

page 2 line 67: (955 U??/L) what "??"do mean?

Author Response

Dear Editor,

We extensively modified the manuscript as detailed below, carefully taking into account the constructive and very useful comments.

1. In this cae report, the author present a patient exhibiting EGPA relapse after Sars-Cov2 vaccination. She previously undewent Covid-19 without clinical  effects. The patient after vaccination was positive for ANCA. Did the patient exhibited anti-IFNalpha antibodies?

We thank the reviewer for her/his questions but in our laboratory is not possible to detect anti INFalpha antibodies, so we did not have this element.

2. page 2 line 67: (955 U??/L) what "??"do mean?

We apologize for these orthographic mistake. We erase the double questions marks.

Round 2

Reviewer 1 Report

The authors have extensively revised their manuscript. No further objections. I would advise cross-checking the last paragraph of the discussion during the proofreading as the English is a little off.

Author Response

Reply to the reviewer 2

The authors have extensively revised their manuscript. No further objections. I would advise cross-checking the last paragraph of the discussion during the proofreading as the English is a little off.

We thank the reviewer for her/his advices. We modified as requested.